# Social Sustainable Education in a Refugee Camp

**Marit Heldal** [1,*] , **Trond Løge Hagen** [2] , **Ingvild Olsen Olaussen** [3] **and Gry Mette D. Haugen** [4]

1 Department of Pedagogy, Queen Maud University-College of Early Childhood Education, 7044 Trondheim, Norway
2 Department of Physical Education, Queen Maud University-College of Early Childhood Education, 7044 Trondheim, Norway; tlh@dmmh.no
3 Department of Drama, Queen Maud University-College of Early Childhood Education, 7044 Trondheim, Norway; ioo@dmmh.no
4 Department of Social Studies, Queen Maud University-College of Early Childhood Education, 7044 Trondheim, Norway; gmdh@dmmh.no
* Correspondence: mhe@dmmh.no; Tel.: +47-9224-9988

**Abstract:** The main objective of this article is to discuss how an Early Childhood Education and Care (ECEC) institution in a refugee camp can promote social sustainable education. By giving empirical examples of innovative pedagogical ideas and practices inside a Greek ECEC institution, this article argues that concepts of formation are ways to promote social sustainable education. The article draws on data from an ECEC institution in which both the children living in a refugee camp and Greek children are located together. With nature as a neutral cultural mediator, serving as a pedagogical framework, children can make new experiences based on participation, equality and mutual respect. Data were produced through field observations, semi-structured interviews and one group interview from March 2019 until September 2019. The empirical data reveal three dimensions that we suggest work as markers for social sustainable pedagogical practice: the importance of nature and play as a facilitator for children's activities; the importance of participation and equality; and the importance of commitment to the community. The findings are discussed in relation with theoretical concepts of formation, with a particular focus on children as active agents and the value of experiences, and the importance of highly qualified educators.

**Keywords:** social sustainability; ECEC institution; nature; child refugee/migrant children; children's participation; play; experiences; formation; reflective pedagogical practice

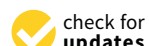



## 1. Introduction and Background

In 2009, the International Journal of Early Childhood (IJEC) published a special issue titled Early Childhood Education for Sustainability (ECEfS), edited by Siraj-Blatchford [1]. Ten years later, this was followed by a new special issue [2], titled Contemporary Research on Early Childhood Education for Sustainability. This focused on the firm belief that early childhood education and care (ECEC) involve the ethical responsibility to care for and work toward sustainable futures for all children. Several researchers suggest that a sustainable future for all human beings and for the world is, to a largely extent, embedded in ECEC [2–6], among others. The International Organization for Migration [7] estimated that globally, there were 244 million international migrants in 2018, making up 3.3% of the world's population. Global displacement was at a record high, with the number of those that were internally displaced at over 40 million and the number of refugees at more than 22 million. Within these figures are millions of vulnerable children struggling to thrive, living large parts of their childhood in refugee camps. This reality has urgent implications for education and child welfare [7] (p. 232).

The main objective of this study was to explore how an ECEC institution in a refugee camp can promote social sustainable education (SSE). With an increase in immigration to Europe, there is a need to enlighten aspects of sustainable education for all children. This

is in line with the UN Sustainable goals (SDG) 4.2 [8], European Governments concerns on challenges that migrants might pose and the OECD's statement that efforts should be made to ensure that the ECEC system becomes more sensitive and responsive to the requirements of immigrant related issues in ECEC education [9,10].

In an official report by UNESCO about the contribution of early childhood education to a sustainable society, Pramling Samuelsson and Kaga [11] emphasize the value of creating spaces for all children to connect with nature, and they highlight the importance of children's agencies to bring about change to support their future lives. The concept of sustainability consists of three interwoven pillars: economic growth, social inclusion, and environmental protection [8]. Whether a society is considered sustainable or not has to do with the quality of life for every member of the community as well as the quality of the community itself. Perception of the content and quality of human welfare varies across different communities and depends on political and cultural divisions, making a clear definition of the concept of social sustainability difficult to establish. Dillard, Dujon and King [12] (p. 4) summarize social sustainability as a process that promotes social health and welfare and is dependent on social institutions that support ecological and economic sustainability, both now and in the future. This is in line with Agenda 21 chapter 36 [13], which points out the importance of people's ability to influence their own life situation and to influence the society in the direction of sustainable development. This is also valid for children living in a refugee camp [14].

Special attention must be paid to children's rights to participate, as stated in the UN goals for a sustainable society [8]. This perspective promotes a conceptualization of children as active subjects and competent beings, an approach that is also central within the sociology of childhood, currently referred to as "childhood studies" [15–23]. This approach—in particular, the understanding of children as individuals with rights, who are equal members of social and cultural groups—is an important perspective in early childhood education for social sustainability [24–26]. Boldermo and Ødegaard [26] (p. 3) discuss the differences and tensions in how the term Education for Sustainability (EFS) is perceived and argue for a turn toward typically human needs such as human rights, democracy, and social issues. Several researchers suggest that education for sustainability can be understood as a process of social and cultural learning that develops new understandings and practices [27–29]. This perspective could also provide better conditions for all children through a value-based approach [26]. This requires a practice in which educators have to constantly reflect on their own values and meet each child as a subject of action and responsibility. According to Pramling Samuelsson and Park [4], such value-based pedagogy should allow children to take initiatives, to think and develop their own reflections. The understanding of social sustainable education in this article is inspired by this value-based approach.

Researchers specializing in this field have indicated a lack of research on migrant children's situations within the context of early childhood education for social sustainability [22,24,30]. In particular, there is a lack of practice-related research that recognizes the Convention on the Rights of the Child [31], where human dignity and education for life is focused within the most formative years [4,26]. This article aims to contribute to fill this gap by illustrating and discussing innovative pedagogical ideas and practices in an ECEC institution for both local and migrant children in a refugee camp in order to promote social sustainable education. Doing research in a refugee camp, a context characterized by political disputes, cultural diversity and where the human needs for social predictability and stability is prominent, could be a "window of opportunity" [32], for the development of a holistic approach to ECEC for sustainability (ECECeF). The research question for this study is: how can educators in an ECEC institution in a refugee camp facilitate social sustainable education for both migrant and local children?

*Theoretical Perspectives*

In order to focus on perspectives concerning social sustainable education in an ECEC institution in a refugee camp, this article underlines the importance of children's formation

processes as socially ubiquitous and continuous, a view that has primarily been dominant in the Nordic ECEC tradition [33–38]. In contrast, the Anglo-Saxon tradition has largely been dominated by a focus on children's learning outcomes and results [39] (p. 19). Huggins and Evans [30] (p. 7) describe different perspectives on both sustainable education and children as active learners. They underline the importance of educators reflecting practices and the need to think consistently about how their chosen approaches are fostering qualities and understandings that are important if children are to act for sustainability. Further, Huggins and Evans [30] (p. 7) highlight the importance of an ECEC practice where educators are supporting children's learning through exploration and experimentation to create their own meanings from processing their experiences. This is also pointed out by Hohr [37] (p. 117), who emphasizes, "Experiencing is something the children do. No one can give them experiences, and the children cannot receive them." This statement acknowledges the child as an active individual and recognizes that the experiences we have are unique. Nevertheless, according to Hohr, we tend to talk about experiences as if they were general. In contrast, he argues that adults should facilitate and be open to discussing the experiences children create, emphasizing the educational advantage in the concept of experience. Experiences depend on the situation in which they arise. Hohr [37] (p. 120) distinguishes between pre-symbolic experience and symbolic experience. Pre-symbolic experiences refer to situations in which children are in immediate interaction with the environment. As children develop their abilities to use symbols (such as language), the ability to communicate symbolically is established and further developed. Hohr [40] also explains feeling as a pre-symbolic mode that goes beyond a particular experience. This is something that educators working with migrant children who have traumatic experiences, have to take into consideration in their everyday pedagogical practice.

A central contribution in the Nordic tradition, related to didactics and curriculum work, is Brostrøm and Hansen [34]. They identify four important aspects of the formation concept: the individual's own activity, equivalent dialogue, feeling of commitment, and action [34] (p. 27). According to Brostrøm and Hansen [34] (p. 29), the individual's own activity is ideal for children to experience themselves as valuable persons with their own opinions and feelings. This is also supported by Hohr [37] (p. 124), who emphasizes play as the child's most important activity, promotes a perspective on play wherein adults must not interfere with the children's projects, and underlines the value of play for its own sake. It follows from this that children's opportunities for play and interaction depend significantly on the educator's pedagogical values and competence, as well as the way they facilitate everyday life in the ECEC institution.

Equivalent dialogue is characterized by a mutual relationship and respect between people, in particular between adults and children [34,41]. According to Løvlie Schibby and Løvlie [42] (p. 19), children who meet adults who are capable of seeing others' perspectives will develop their own ability to be part of equivalent dialogues. Equivalent dialogue is an important premise for equivalent understanding, which in turn is an important premise for social sustainability. Through the individual's own activities and their participation in equivalent dialogues, both children and adults may generate new perspectives and a sense of feeling commitment to their surroundings [34]. According to Brostrøm and Hansen [34] (p. 32), there is no formation before knowledge is transformed into action. When children get the opportunity to be active agents and take part in their environments, their actions influence the culture and surroundings. Consequently, children experience being active contributors in the community. In this way, children might also acquire new understandings and attitudes towards social sustainability as pointed out by Hedefalk [43]. This is in line with Davis' [44] (p. 25) multidimensional approach to children's rights for ECECfS, basic rights as promulgated in the United Nations Convention on the Rights of the Child, children's agentic participation rights, collective rights, intergenerational rights and biocentric rights. These rights deal with important aspects for educators bringing people together to serve collective interests, deal with responsibility between generations and that all biological creatures have value and inherent rights to life.

Furthermore, we argue that the pedagogical ideas referred to above are also laid out in the work of Chan et al. [45] (pp. 37–38). They suggest that the Chinese character He or "harmony" might serve as a theoretical basis for interpreting the concept of sustainable development, by distinguishing harmony within the individual, harmony among individuals and harmony between humans and nature.

In recent years, the use of nature as part of the educational environment in ECEC has become more prominent (for an overview, see [46,47]). Studies show that interaction with nature environments facilitates a dynamic, social, all-consuming, and creative play [48–51]. Using nature as an educational environment provides possibilities for children and for educators to enrich play, because nature provides "tools", "toys", and a variety of spaces in which to play [48,52,53]. Nature is also, in many ways, genderless, and children's ways of playing in nature are strongly similar around the world [54,55]. Play is important for children's formation [56]. Through play, children create new experiences, express themselves, and interact with each other. Nature facilitates play based on inherent affordances [57] in the environment and may function as a socially sustainable environment for children's formation that transcends age, gender, and culture [54,58].

Several of the children who are part of the research in this article carry difficult experiences, such as traumas from fleeing wars and separation from family. However, Steinsholt [59] highlights that the negative embedded in real experiences does not only have a negative effect; by remaining open to new experiences, children are able to re-evaluate and transform what they know and become something more [59] (p. 108). To create a safe space where children feel invited to create new experiences is one of the greatest challenges for ECEC institutions, in particular for children living in refugee camps.

Huggins and Evans [30] (p. 5) argue that in a global context of accelerating change, educators cannot afford to be rooted in the past, or bound by taken-for-granted practices embedded in their own local experiences and traditions. In order to underpin sustainability in the ECEC field, there need to be significant shifts in many practitioners' mind-sets, involving them in critical reflections upon existing values and beliefs. According to Huggins and Evans [30], this can be done by giving educators opportunities for safe dialogical spaces where they can develop a new understanding of how ECECfS can enhance their practice and benefit children's lives.

Biesta [36] states that pedagogical wisdom is not a specific quality or skill but a constant reflection on one's own experiences in practice, and thus it is relevant to the discussion in this article on how educators emphasize children's experiences, play and activities.

Biesta [36] (p. 4) also describes a political situation consisting of a misguided impatience that pushes education in a direction where children are being made to fit the educational system and where education is about a perfect match between "input" and "output". He questions where the causes of this unfortunate state of affairs lie, and thus whether society or the child needs adaptation. His answer lies in what he calls "the educational way", where education is an encounter between human beings and where children are seen as subjects of action and responsibility. These pedagogical perspectives are relevant when discussing pedagogical practices and how an ECEC institution may promote elements of socially sustainable education.

## 2. Methods

This article draws on data from one ECEC institution in a refugee camp. Data include 14 field observations, 36 semi-structured interviews and one group interview (see Table 1). With regard to the research question, combining these approaches gave the opportunity to follow the participants over a longer period, observing them in their natural setting.

**Table 1.** Data.

| Empirical Material | N | Participants | Duration | Project Plan |
|---|---|---|---|---|
| Semi-structured interviews—phase 1 | 30 | Three Greek educators, one man and two women | Approximately 15 h (1 pr month à 30 min) | October 2017–March 2019 |
| Participatory observation | 14 | 16 children age 2.5–6 years. 8 children from Greece and 8 children from respectively: Syria, Afghanistan, Iraq, and Kurdistan. | 305 min Each observation 20–25 min | March 17th–March 21th 2019 |
| Semi-structured interviews—phase 2 | 6 | Three Greek educators, one man and two women | Approximately 3 h (1 pr month à 30 min) | March 2019–Sept. 2019 |
| Group interview | 1 | Three Greek educators, one man and two women | 75 min | Sept. 11th 2019 |

The field observations were implemented by using participant observation [60]. Schensul, Schensul, and LeCompte [61]) (p. 91) define participant observation as "the process of learning through exposure to or involvement in the day-to-day or routine activities of participants in the researcher setting". In this study, the participant observation enabled the researcher to learn about the everyday activities in the ECEC institution in their natural setting. The field observations together with the semi-structured interviews gave important knowledge to prepare for the group interview.

A semi-structured interview is a verbal interchange where the researcher attempts to elicit information from the participants [62]. By asking questions unfolded in a conversational manner, the participants were given the chance to explore issues important to them. In this study, semi-structured interviews were produced over two periods (see Table 1). Phase one was conducted by one of the researchers and was used as a way of "finding a focus and knowing where you stand". According to Mason [63] (p. 13), this is an important, but often ignored, challenge that qualitative researchers should deal with in order to "design an effective project with a clear and intellectually worthwhile focus to explore your topic". Because the ECEC institution is located in a refugee camp, localized in an area affected by challenges related to the "migrant crises", this phase was particular important in order to position the project within the field of ECEC for sustainability. The second phase of the semi-structured interviews between the researchers and the three educators took place after the observation, with the purpose to prepare for the group interview.

A group interview was chosen as one of the approaches because it is suitable as a more efficient use of resources, and as a means of adding valuable insights to questions addressed in this research [64]. By using this approach, the research group had the chance to meet the participants in an informal setting in the ECEC institution. The semi-structured interview was conducted, allowing the participants to explore the questions from multiple angles [62,64]. While field observations are often enveloped in interpretations, the group interview provided the research group with complementary information that came directly from the educators [63]. Participants with experiences related to the same phenomenon could also activate each other's reflections [64].

### 2.1. Physical Context

The study was implemented in an ECEC institution located in a wooded area in a refugee camp on the island of Lesvos, Greece. The institution was established in 2017, and one of the main goals was to include both Greek children and children from the refugee camp. The entire outdoor area of the ECEC institution is a nature playground. There are no residential buildings, although a tent is provided for use on cold days. There is also no access to the sewage system, so composting toilets have been built. The water comes from a tap behind the compost building. An outdoor kitchen includes wooden logs to sit on and a large wooden table where the children can have meals and gatherings. The

nature environment is organized into different spaces. For example, one space is designed for aesthetic activities like painting and construction; one is for climbing, and another for physical activities. Quiet places are available where children can relax, tell stories, or read. The basic idea is that the environment facilitates possibilities for a large variety of experiences, play, and activities that are important to children's formation.

### 2.2. Research Group

The research group consists of four researchers who have been involved with the ECEC institution in various ways. One of the researchers founded the ECEC institution, together with parents and educators from the local community. This researcher has worked and lived in the refugee camp where the ECEC institution is located at different periods (for a total of seven months between January 2017 and October 2020). The experiences and knowledge have provided the research group with important background information about the complexity of the ECEC institution and the interplay among the educators, the children, and their families.

### 2.3. Participants

At the time the observations and the group interview were conducted, 16 children and three educators (Maria, Elli, and Giorgos) were affiliated with the ECEC institution. Eight of these children were from the camp itself, and eight were from the local community of Lesvos. The children from the refugee camp had fled their home country and had been exposed to exploitation or risk, which can lead to enduring trauma. Thus, this study meets Mason's [63] demand of empirical significance with reference to the global challenges regarding migration (often referred to as the migrant crisis) and the number of refugee children on the move.

The participants in this study were selected by strategic sampling [63] (p. 124) because of the relevance to the research question based on their work as educators in the ECEC institution in the refugee camp. Detailed sociodemographic data of the participants are provided below (see Table 2).

**Table 2.** Participants' sociodemographic data.

| Fictive Name | Gender | Age | Educational Background | Current Posisiton | Previous Position | Years of Teaching Experience |
|---|---|---|---|---|---|---|
| Maria | F | 32 | Master in childhood studies and legal rights. | ECEC educator in a ECEC institution in a refugee camp | ECEC educator in a forest school in Germany Field worker in a protection unit in a refugee camp | 5 |
| Elli | F | 26 | Bachelor, ECEC studies | ECEC educator in a ECEC institution in a refugee camp | ECEC educator in a forest school in Spain | 3 |
| Giorgios | M | 32 | Bachelor, Elementary school teacher | ECEC educator in a ECEC institution in a refugee camp | Elementary school teacher | 10 |

### 2.4. Procedure

This study includes three different methods when it comes to data collection: field observation, semi-structured interviews and a group interview.

#### 2.4.1. Participant Observation

Fourteen field observations were conducted by one of the researchers over a period of four days in March 2019 (18.3–21.3) by using participant observations. This technique

allowed the researcher to make observations as a natural participant in the everyday activities mentioned below. Before conducting the field observations, an observation chart was prepared. This chart consisted of two columns marked "descriptions" and "interpretation", with the date and time above. The field observations were structured because the researcher had clarified, in advance, which situations were to be observed. These situations were: arrival, free play, activity related to the meal (preparation, hand washing, the meal itself and cleaning up) and the end of each day where everyone would gather in a common circle. The role of the researcher was to observe pedagogical practices in the ECEC institution's everyday work using participant observation [60]. Each observation period lasted twenty to twenty-five minutes and provided a lens into the daily routines and everyday pedagogical activities.

### 2.4.2. Semi-Structured Interviews

During the project period, 36 semi-structured interviews were an important source of information. Thirty of these interviews were conducted in phase one and six in phase two (see Table 1). The semi-structured interviews took place as dialogues. Dialogues can be understood as a way to "jointly create meaning and shared understanding" [65] (p. 814), and were produced both through face-to-face communication and digital meetings between the researchers and the educators. Each semi-structured interview was based on a specific topic related to the research question (e.g., daily routines, pedagogical practices, pedagogical values).

### 2.4.3. Group Interview

A group interview with the three educators (see Table 1) took place at the location of the ECEC institution by the research group on 11 September 2019. The interview lasted for 1 h and 15 min and was conducted in English. The session was audio recorded and transcribed verbatim. Throughout the group interview, one of the project's researchers acted as a moderator for the group interview's main theme, addressing educational perspectives and practical pedagogical examples by asking about their fundamental pedagogical ideas and routines, their experiences from pedagogical work with the children, intervening with the parents, cultural challenges with gender equality as well as how they experienced nature as a venue for pedagogical practice. The participants reflected on these topics with each other and responded to additional questions from the research group. In this way, the group interview provided nuanced and in-depth descriptions of the educators' experiences.

### 2.5. Data Analysis

The data from the field observations, the semi-structured interviews and the focus group interview were initially analyzed separately [66]. Looking at the written texts of the sources, the researchers conducted an individual open coding process [67], producing a large number of descriptive codes (e.g., showing love, showing respect, bodily expression, and different kinds of daily activities). In the next phase, the research group worked together in a three-step process that was conducted through several analysis seminars. First, the group compared and discussed the open coding related to the field observations, semi-structured interviews and the group interview. Second, the group conducted a cross-sectional analysis [63] where the different sources of data were seen in relation to each other; here, the data were merged into 21 different categories (e.g., caregiving and caregiving experiences, participation, coexistence, vocal and body expression, and nature). Third, the analyses were conducted through axial coding, a set of procedures in which connections were made between the codes involving conditions, context, and consequences [67] (p. 96). Through this process, the empirical material was organized into three main dimensions: (i) the importance of nature and play as facilitators for children's activities; (ii) the importance of participation and equality; and (iii) the importance of commitment to the community. The three dimensions are presented and discussed in the "Findings and discussions" section.

*2.6. Ethical Considerations*

The Norwegian ethical guidelines for research in social sciences [68] were followed in all stages of this study. The participants were informed about the study and their right to withdraw from it at any time. For the children who were observed, we received informed consent from their parents. Handwritten notes were taken throughout the day, and all personal information was anonymized in the written material.

Children in refugee camps are exposed to an influx of multiple adults with different motives, such as journalists and volunteers who do not respect the children's need for privacy and their intimate zone. Showing special consideration to children is important in all research, and particularly in research with children who have had traumatic experiences [69,70]. This was the case in our research and was a reason to take ethical considerations extra seriously. The material describes general examples that cannot be associated with a particular child. The children, their nationality, and other ethnic markers cannot be identified in the data material. A sustainable pedagogy stems from the challenges associated with this particular ECEC institution. The children and adults in this context are sometimes in a vulnerable situation. However, in this article, we focus on describing the sustainable pedagogical practice in this ECEC institution.

## 3. Findings and Discussion

This section is organized into three parts presenting the three dimensions that were revealed from the analysis. The first part reveals the importance of nature and play as facilitators for children's activities. The second part discusses how children's participation in daily routines and a focus on equality are important in the ECEC institution's daily life. Finally, the third part discusses the importance of being part of a group and of experiencing commitment to the community (see Table 3, main results and data interpretation).

**Table 3.** Main results and data interpretation.

| Formation as the Core Concept | | | |
|---|---|---|---|
| **The Importance of Nature and Play** | **The Importance of Participation and Equity** | **The Importance of Commitment and Community** | **Three Main Dimensions** |
| 1. "The oldest children can, for example, climb over obstacles, and the youngest one can crawl under. In that way, they learn from each other, and everyone finds their own ways to challenge themselves and to explore the environment" (Elli).<br><br>2. "My opinion is that what calmed her down was this place...nature. She could have her own space; it was something she needed." | 1. "The educator (Elli) takes the child's hand, and together they walk towards the shelter in order to go and get the dog's food, while the dog comes running after" (observation log no. 5).<br><br>2. The children are "feeding" the insects while they are studying their moves and reactions" (observation log no. 2). | 1. " . . . and if we encourage the child to help another child in a difficult moment, and if we learn how to share. And every day I see the scene changing. I feel proud" (Maria).<br><br>2. "We are equal in nature. And, of course, there are no toys that are made specifically for girls or boys, and there are no colors labeled for girls or boys" (Elli). | **Data** |
| 1. The nature environment also offers the children rich opportunities performing activities over time, they develop language skills and learn to communicate in spite of cultural challenges and language barriers.<br><br>2. An active approach towards nature, in combination with individual adjustment and facilitating, helped this girl to communicate and receive experiences that led to new understandings and attitudes both within herself and towards others. | 1. By implementing daily routines based on children's participation and mutual respect, the educators have experienced a change in the children's behavior towards animals.<br><br>2. According to the educators, this is a change in attitudes, because at an earlier phase the children used to kill insects. However, the educators have been clear about focusing on the value of life within nature. | 1. Providing the children with skills that give them opportunities to act as committed citizens and be part of the society, is something the educators emphasize.<br><br>2. The environment promotes equality and creates a platform for dynamic play across culture and gender. | **Findings** |

### 3.1. The Importance of Nature and Play as Facilitators for Children's Activities

The close relation to nature permeates the data material in the sense that nature offers a diverse and open educational environment, serving as a prerequisite for the pedagogical practices and children's experiences. Working with children who have had traumatic experiences related to war, involuntary separation from relatives, and unfit living conditions in different refugee camps requires a particular awareness around how the pedagogy is practiced in order to promote socially sustainable education. Despite their different professional backgrounds and varied practical experiences, the three educators emphasized the value of creating a space where children have the opportunity to create experiences through activities with peers and to be agents in their own lives. When we asked the educators to reflect on their pedagogical practices in relation to the fact that the ECEC institution was located in a refugee camp and that the group of children was diverse (including both children from the refugee camp and local Greek children), they acknowledged that the process of developing good pedagogical practices had not been straightforward:

> "In the beginning, we came with the romantic idea that children would play with each other, because culture and language don't matter—we are all together. But this was not the case. The group was very segregated. We had to change our practices several times and see how we could bring the families and children together."
>
> (Maria)

However, from the very beginning, the educators focused on the importance of free play. By organizing the outdoor environment into different zones, they gave the children opportunities both to find a personal space where they could engage in their own individual activities and to explore the environment, either alone or together with peers.

> "A typical day offers plenty of time for free play. We observe that, as months go by, children get increasingly confident in managing their own time. Their need and desire for adult support in this diminishes."
>
> (Maria)

This pedagogical practice underlines the importance of providing opportunities for children to have new experiences [37,41,42]. The children who are newcomers in the community do not speak Greek or English, and although the educators have learned some words in Farsi, Arabic, and Kurdish, verbal communication is challenging, and thus they need non-verbal expressions and understanding. The educators pointed that they are dedicated to giving the children attention, both verbal and non-verbal:

> "We do not ignore the children that pursue our attention. If we cannot interact with a child at the specific moment, we take a moment to explain ourselves ( . . . ) We are constantly looking for suitable ways to communicate, verbally or otherwise, with each child. All children are capable of expressing their views when enough effort is put into choosing appropriate ways of communicating with them."
>
> (Maria)

The educators further shared that the nature surrounding the institution becomes an important tool/factor in facilitating the children's play. Since nature is so diverse and rich in affordances [57], the environment offers challenges and possibilities for all the children regardless of their age, gender, and cultural background. Birkeland and Grindheim [27] (p. 9) also aim at providing surroundings that stimulate positive interactions. According to the educators, the children use their creativity to explore nature on their own terms, creating dynamic and inclusive play:

> "The oldest children can, for example, climb over obstacles, and the youngest one can crawl under. In that way, they learn from each other and everyone finds their own ways to challenge themselves and to explore the environment."
>
> (Elli)

In this way, the nature environment also offers children rich opportunities for presymbolic experiences [37] (p. 120), and by performing activities over time, they develop language skills and learn to communicate in spite of cultural challenges and language barriers. This practice might be seen as being in line with what Hohr [37] (p. 120) labels as an educational advantage in the concept of experience. The nature environment also seems to be critically important in relation to the way the educators work with children who have experienced trauma before they enter the ECEC institution.

By giving the children time to feel comfortable and to trust the adults, the educators were open to making individual adjustments for each child, as illustrated in an educator's story about one traumatized girl who came to the ECEC institution and who needed time to adjust:

> "My opinion is that what calmed her down was this place...nature. She could have her own space; it was something she needed. If you have a child like this and you put him or her with 20 other children in a small space, he or she can't find a way to be alone (...) When she finally found that "now I can participate in things", all the space, all the people were open for her to do it. So that is one reason that I am saying that the forest, the big place, was something that indulged her . . . and made her feel better."
>
> (Giorgos)

By organizing the physical environment and ensuring time for play and "ways to be alone" the educators acknowledge several of the articles in the UNCRC regarding children's right to play and participate as well as children's right to be cared for, all important factors in the making of a socially sustainable educational practice. According to Brostrøm and Hansen [34] (p. 29) play for play itself, is especially important because it is through play that children reflect on their experiences, culture and reality, transforming them into a medium that is clearer and more understandable [37]. Our findings reveal that facilitating play in nature environments contributes to children's active learning through experiences. Huggins and Evans [30] highlight the importance of an ECEC practice that envisages children as active in their learning processes through exploration and experimentation to create their own meaning from processing their experiences.

### 3.2. The Importance of Participation and Equality

At the same time as the educators emphasize free play and exploration, they also underline the importance of children's participation through the daily routines in the ECEC institution. It is emphasized that everyone, both children and adults, should contribute within the community, and dimensions such as equality and participation are expressed:

> "We are strict as long as the function of the community and the equality in it is concerned. It is one of the baselines. We share with children responsibilities that concern the community We have observed that children are not only capable of undertaking responsibilities, but also happy to do so."
>
> (Maria)

Both the observations and the group interview provided several examples of how such activities were conducted. Activities like sharing food, telling everyone hello and goodbye, and showing mutual respect affected the children's practices in terms of how they reacted, toward each other, nature, animals, and adults. A few years ago, the ECEC institution adopted a stray dog that now lives in the camp. One daily routine is that the first child who arrives in the morning is responsible for giving the dog food and water.

This morning routine was observed several times, with different children being given the responsibility to feed the dog:

> "The educator (Elli) takes the child's hand, and together they walk towards the shelter in order to go and get the dog's food, while the dog comes running after."

> (observation log no. 5)

Taking care of the stray dog gives the children experiences related to participation, responsibility, and solidarity, which according to Broström and Hansen [34] are key factors in the children's formation and important factors in a social sustainable education. This attitude was discerned in the following observation:

> "One of the children [from the camp] is standing alone on a hill, while one of the others [one of the Greek children] comes running towards him. Standing side by side they both look down—something is happening on the ground. One of them [from the camp] suddenly runs down to the "food place". He catches a piece of bread and runs back to his friend. The children are "feeding" the insects while they are studying their moves and reactions."

> (observation log no. 2)

It is interesting to observe that the children collected and observed the insects, giving them food and water. According to the educators, this is a change in attitudes, because at an earlier phase the children used to kill insects. However, the educators have been clear about focusing on the value of life within nature. The children's interest for the insects also functions as a catalyst for play or a pre-symbolic experience (to use Hohr's [37] term) between children who do not speak the same language. By giving children the opportunity to perform activities that both migrants and local children can engage in despite linguistic and cultural obstacles, new experiences are created. This "exploring attitude" toward nature and insects is a typical result of the educators' pedagogical practices, which, as illustrated in this example, focused on exploring processes rather than learning outcomes (e.g., learning insect names). According to the educators, this "sustainable attitude" emerged during the children's time in the institution. In this way, the practice underlines Hohr's [37] (p. 17) argument that given the right circumstances, children develop emotions towards the reality they are a part of and create room for new experiences, as exemplified by caring for animals and insects and making new relationships.

In order to work with a pedagogy based on social sustainable education, the educators meet every afternoon in what they call "the reflection circle", in which they discuss their experiences and observations throughout the day. For example, during these reflections, they discuss in detail how they should interact with the children in different situations in order to facilitate participation and equality, as described below. This reflection practice is in line with what Biesta calls "pedagogical wisdom" [36] and further illustrates how the educators emphasize that the experiences we have are unique [37]. Wolff and Ehrstrøm [28] (p. 16) strive to awaken curiosity as well as active self-criticism. The educators emphasize children's individual activities and encourage equivalent dialogue and interaction among them. "For example, when a child asks us for some help or information that another child could provide them with, we encourage them to address their peer" (Maria). In interactions with the children, the educators are careful about recognizing and respecting every child's private space.

> "We recognize that we do not have the right to step into the child's private space without his or her consent, and instead of taking a child's hand in ours, we propose our hand to the child and ask if they want to hold it. Instead of hugging a child, we open our arms for a hug so that the child gets the chance to decide if they want to enter."

> (Maria)

This example vividly illustrates how small nuances and small details in the pedagogical practice are significantly important in the pedagogical "doing". When the educators

reflect on their interactions with the children and are aware of children's private space, they act in line with Brostrøm and Hansen's [34] concept of equivalent dialogue and Biesta's [36] concept of pedagogical wisdom, which both are characterized by a mutual relationship and respect between children and adults.

The educators further underlined that activities like those described in this section that emphasize equality and mutual respect led to more harmonic interactions and more relaxed play among the children and between the children and the adults. This is interesting, as it might correspond with Chan et al. [45] (p. 37) and their concept of "harmony" as a theoretical basis for interpreting the concept of sustainable development.

*3.3. The Importance of Commitment to the Community*

The fact that the ECEC institution includes both local children and children from the refugee camp is viewed as both a strength and a challenge. The choice was made consciously as a way to seriously address the challenges associated with living with diversity. It is also a way to look beyond the understanding of 'us' versus 'them'. The educators expressed that they invest in cultural diversity as a way to build what they call a fair society that respects nature and human beings. Marouli [29] (p. 13) states that pedagogy today should respect and build on cultural diversity. According to the educators, one of the challenges was that they always have to consider the way they interact with the children, in order not to insult them or their families: "If we only had local children and no one from the refugee camp, we would allow more things. Now, we are careful so as not to insult some cultural aspects" (Maria). The blended group of children can also serve as a reason for letting the children attend the institution. At times, parents from the refugee camp question the outdoor space because some of them are not used to ECEC institutions in general, and outdoor pedagogy in particular. However, the educators stated:

> "When we tell the parents in the camp that local people want their children to come here ( . . . ) then it changes their minds, or something."
>
> (Elli)

In addition, the educators highlighted the nature environment as a gender-neutral and inclusive environment.

> "It is nature that is there for all of us, and all the resources are the same. There is a social distinction, but it simply disappears because...we are equal in nature. And, of course, there are no toys that are made specifically for girls or boys, and there are no colors labeled for girls or boys."
>
> (Elli)

The children are allowed to intuitively explore and engage in play based on their own subjective perceptions of the environment, indicating that nature—with its rich and diverse affordances—creates a platform for dynamic play across culture and gender [51,54,57]. As shown in the quotation above, the environment promotes equality. Buildings, interiors, books, and toys are coded as a culture or a nationality, while nature is more or less played with in the same way throughout the world, and the educators experience it as a unifying environment where children easily play together, learn about each other, and become friends [54,55].

The importance of gender equality is underlined in the Agenda for Sustainable Development [8] (p. 19), Goal 5. Even if nature, to a large extent, facilitated equality and equivalent dialogue, the educators also met challenges. For example, one of the children did not want to do the dishes, which was one of the obligations the educators expected the children to participate in. The child explained that "because his father didn't do it, he shouldn't either" (Giorgos). The educators related this to cultural factors and traditions while also underlining the importance of promoting gender equality in the ways they talk and act. They emphasized their position as (gender) role models, who build, cook, and participate in various chores regardless of gender. This awareness is also pointed out by Løvlie Schibby and Løvlie [42] (p. 19), who state that a child who meets adults who

are capable of taking others' perspectives will develop their own ability to adopt other perspectives. This actually happened with the boy that did not want to do the dishes. Despite his attitude, the educator's strong emphasis on equality and commitment to the community challenged the boy's attitude, and after a while, he started to contribute:

> "Finally he did it … and he was very excited about it. He cleaned everything up and he washed the table."
>
> (Giorgos)

This example can be viewed in light of Brostrøm and Hansen's [34] (p. 30) concept of equivalent dialogue, which highlights how the reflection process initiated between two individuals changes and allows for development in both. Through an assessment of one's own values, new insights are established on a deeper level that make one feel responsible and therefore take responsibility. In their review of social education, Boldermo and Ødegaard [26] (p. 4) state that "the premises for social inclusion and belonging can be subject to negotiations", often with age and gender working as excluding factors. In the context of the ECEC sector, it is critically important that educators are aware of such challenges and apply pedagogical practices that acknowledge children's own activities, equality, and commitment to the community, as illustrated in the example above. By being exposed to such a pedagogical practice, the boy in the example changed his attitude and started to act differently. By washing the dishes, he gained new experiences based on equality and commitment to the community, important pieces in the making of a sustainable society.

The educators highlight the importance of being aware of the children and taking the children's perspectives. The examples above illustrate in different ways how nuances and small details are significant in the pedagogical "doing". They underline that this is a difficult but not impossible process, which, in subsequent steps, might lead to a society based on values of social sustainability:

> "Everyday there are examples that show us that there are ways to achieve this. And not very sophisticated and complicated scientific ways, but ways in how we talk to each other, and if we encourage the child to help another child in a difficult moment, and if we learn how to share. And every day I see the scene changing. I feel proud."
>
> (Maria)

By learning daily routines and caregiving skills that produce positive practices and attitudes, the children will take advantage of these experiences in their everyday life, now and in the future. The educators explained that they try to make the transition from the ECEC institution to the school as smooth as possible; however, some particular challenges might be faced by the children from the camp, such as their parents not speaking the local language. However, they find that the children who have spent substantial time in the ECEC institution manage the transition quite well, as illustrated in the example below, where one of the educators described a conversation with a school teacher one month after school had started:

> "The teacher told me that during food eating times and the preparation, she/he does everything ( … ), knowing how to treat her/himself and treat his/her things and food and the table, and he/she always helps out with this don't have this culture (...) And I wanted to give her [the teacher] the hint that if she/he [the child] is hugged, then she/he is relaxed and can come and talk. And the teacher told me, 'I know, because he/she asked for it him/herself from the first day, and showed me that this is the way to have him/her calm."
>
> (Maria)

According to Pramling Samuelson and Kaga [11], it is important that education for sustainable development starts in early childhood, because values, attitudes, and behaviors developed in early childhood have long-lasting impacts. This is in line with

Davis' [44] multidimensional approach to children's rights, in particular collective rights and intergenerational rights, which means bringing people together to serve collective interests, and dealing with the responsibility between generations. According to Korsgård, Slagstad and Løvlie [71], formation is about humanity, democracy, and solidarity, but also about the responsibility of each individual to realize and put into practice their own self-determination. By playing an active role in their own lives, the children attending the ECEC institution, regardless of background, are given the opportunity to create experiences that make them ready to "feel home in the world", as worded by Biesta [36].

## 4. Concluding Discussion: Social Sustainable Education Now and in the Future

In order to explore how an ECEC institution in a refugee camp can promote social sustainable education (SSE) the analysis revealed three empirical dimensions. First, nature and play as facilitators for children's activities; in this dimension, the pedagogical practice described in the data showed how nature served as a mediator that facilitated children's activities, play and pre-symbolic experiences. Second, the importance of participation and equality; here, participation through daily routines, mutual respect towards human, animal and nature, equivalent dialogues, as well as pedagogical reflections among the educators was pointed out as pivotal factors. Third, the importance of commitment to the community; in the final dimension, the educators invested in cultural diversity and nature as an inclusive environment and emphasized the commitment to the community.

The perspective of formation is cutting through the three empirical dimensions, which we suggest is a key element in promoting social sustainable education. Brostrøm and Hansen [34] (p. 32) argue that no formation occurs before knowledge is transformed into action. When both educators and children have the opportunity to be active agents and take part in their environment, processes of formation occur. With support from the present research [4,26,30,38] this article shows how an educator's reflection upon existing values and beliefs in a context characterized by upheavals due to the "migrant crises" can be an example of social sustainable education for both local and migrant children.

As underlined in "The 2030 agenda for sustainable development" [8] (p. 19), the SDGs include a focus on high-quality education for all children worldwide. According to Goal 4.2, by 2030 all girls and boys should have access to high quality early childhood development, care and pre-primary education so that they are ready for primary education [8] (p. 19). Given that a large part of the world's population now and in the future will be migrant children, the pedagogical examples presented in this article are crucially important to show how educators in an ECEC institution in a refugee camp can promote social sustainable education for all children.

Migrant children are always at risk of being perceived as outsiders in a community to which they do not belong, as might be the case for the migrant children in Lesvos as well as for other children with migrant backgrounds around the world. Segregation is a potential threat to social sustainable education. One of the choices the educators made to facilitate a social sustainable education was to bring together local and migrant children. The diverse group of children offers opportunities for different experiences to be acknowledged and new experiences to occur. This idea tries to anchor the global influx in the local and to take seriously the challenge to provide better conditions for all children [26]. This practice, and the thinking associated with it, work against the perception of "us" versus "them".

According to Boldermo and Ødegaard [26] (p. 9), one dominant optimistic, future-oriented path in the field of social sustainability considers children problem-solvers. They question whether this approach gives too much credit to the child's competence and thus requires too much responsibility "for children to bear on their own" (Boldermo and Ødegaard [26] (p. 10). While we do not reject this argumentation, we do not entirely rely on it, because as Biesta [36] (p. 117) points out, "Whereas children can never be ready for political existence they also always have to be ready for it."

The children in the ECEC institution presented in this study have different experiences connected to political realities. Those from the camp have fled from war, lived in transit,

and experienced instability and unpredictability, while the local children have experienced social changes due to the migrant crisis. Common to all of the children was the fact that they did not choose to be part of a particular political picture or local context, yet, as shown in this article, they have to deal with the situation. By establishing an ECEC institution in a refugee camp with educators emphasizing a pedagogical practice based on play, dialogue, and experiences, the children might acquire new experiences, be active contributors in the community and develop new understandings. The pedagogical practice from this ECEC institution may contribute to discussions and further reflections on how educators can facilitate good social sustainable education for all children.

## 5. Limitations of the Study and Future Directions

Drawing on a limited qualitative sample, the findings presented in this article cannot be generalized beyond this particular research setting.

However, according to Miles and Huberman [66], qualitative data sampling and analysis is driven by a conceptual question more than the concern for representativeness, and the prime concern is with the conditions under which the research is produced [66]. By doing research in an ECEC institution in a refugee camp, this article aims to advance reflections into new areas of social sustainable education that are, so far, relatively unexplored [26].

Several researchers have pointed out that a foundation for sustainable understanding is shaped in early childhood [1,11,24], making quality ECEC necessary for all children. Challenges caused by migration, such as refugee camps and children's fluctuating environments, highlight the need for social sustainable education, and Boldermo and Ødegaard [26] have called for practice-oriented research where human dignity and education for life should be motivating forces. This article attempts to meet the request for practice-oriented research by presenting new ideas of ECEC pedagogical practice, in which one of the key ideas is to bring migrant and local children together and acknowledge nature as an important pedagogical environment. As Marouli [29] points out, today's pedagogy should involve the community of learners in the creation of a new philosophy of life, in which cultural diversity is important. In line with Wolff and Ehrström [28] who argue that it is possible to implement social sustainability in higher education, we have showed that it is possible to implement social sustainability in an ECEC institution. With nature as a surrounding environment and highly qualified educators, an emphasis on participation, equality and commitment to the community may serve as directions for both ECEC and early childhood teachers' education for social sustainability, now and for the future.

**Author Contributions:** Conceptualization, M.H.; methodology, M.H., T.L.H., I.O.O. and G.M.D.H.; validation, M.H., T.L.H., I.O.O. and G.M.D.H.; formal analysis, M.H., T.L.H., I.O.O. and G.M.D.H.; investigation, M.H., T.L.H., I.O.O. and G.M.D.H.; resources, M.H., T.L.H., I.O.O. and G.M.D.H.; writing—original draft preparation, M.H., T.L.H., I.O.O. and G.M.D.H.; writing—review and editing, M.H., T.L.H., I.O.O. and G.M.D.H.; visualization, M.H., T.L.H., I.O.O. and G.M.D.H.; supervision, M.H., T.L.H., I.O.O. and G.M.D.H.; project administration, M.H., T.L.H., I.O.O. and G.M.D.H.; All authors have read and agreed to the published version of the manuscript.

**Funding:** This research received no external funding.

**Institutional Review Board Statement:** The study was conducted according to the guidelines of the The Norwegian ethical guidelines for research in social sciences [68].

**Informed Consent Statement:** Informed consent was obtained from all subjects involved in the study.

**Data Availability Statement:** The data presented in this study are available on request from the corresponding author. The data are not publicly available due to ethical reasons.

**Conflicts of Interest:** The authors declare no conflict of interest.

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
