# Peer review of "Social Sustainable Education in a Refugee Camp"

_sustainability, doi:10.3390/su13073925_

Round 1

Reviewer 1 Report

Dear authors,

The submitted manuscript has very good intentions and although it is a topic of interest to the scientific community, I observe some weaknesses. I comment on them below:
-The qualitative methodology used does not have sufficient consistency to be published in a high impact journal.
- The discussion and conclusions are not related to the main objective.
-Some questions to answer are: What is the main question addressed by the research? Is it relevant and interesting?
What does it bring to the subject area compared to other published materials?
-Are the conclusions consistent with the evidence and arguments presented? Do they address the main question posed?
- Connect the conclusions in relation to the results obtained so that they present greater forcefulness.
-Indicate how the results obtained are related or not to the literature and the opinion of prominent authors on the subject.
- They should include tables and graphs to facilitate reading for readers.
- Discussions that confront the results obtained with the experts in the field should be included.
- It is recommended to update the bibliography

Author Response

Thank you for in-depth and constructive comments. Attatached you will find our response.

Best regards

Reviewer 2 Report

Although you do a narrative article, you must specifically define the sample, (number, age, sex, ethnicity) of the refugees.

The content is well described and contextualized, with an adequate background to contextualize the study.

The research design, questions, hypotheses and methods are well presented and developed, meeting the standards of the journal.

The article is duly referenced, with current references that adequately contextualize the state of the question.

The discussion is coherent, but in the absence of quantitative data it cannot be properly objectified.

The biggest aspect to improve is to specifically define the sample and present the data clearly, not just narratively. It is necessary to present quantitative results and make use of tables or graphs that evidence or support the conclusions.

The conclusions must be improved. The results obtained must be clearly stated. The conclusions should be much more summarized and be consistent with the results obtained.

I suggest indicating proposals for improvement to the participants.

Author Response

Thank you for in-depth and constructive comments. Attached to this document is our comments on how we have re arranged the article.

Best regards

Reviewer 3 Report

The research problem is of special relevance because it constitutes a contribution to the knowledge about the educational actions developed in refugee camps and their relationship with its immediate environment. Its connection with sustainable education is adequately justified. To this end, a qualitative methodology is applied through interviews and participant observations. The results reveal which pedagogical facilitators could promote a sustainable social education in these contexts.

The following corrections and clarifications are suggested to improve the quality of the manuscript:

1. Please, reorganize the information contained in section 2 (line 167 and following) according to international scientific standards: participants, techniques applied (including evidence of its validation), design and procedure, and data analysis (referred to here as "The analysis processes").

2. Please, make a methodological distinction between "interview" and "focus group" (line 208 and following) . Although these are two qualitative techniques, they have specific characteristics and procedures.

3. Write, in a differentiated and detailed way, a section that discusses the results in the light of the most recent results offered by other studies (this section could be based on what is included in section 4), and a section with your conclusions (lines 525 and following).

4. Write a section with the limitations of the study.

Author Response

Thank you for your in-depth and constructive comments on the article.

Attached is our comments on how we have rearranged the article in line with the feedback given.

Best regards

Round 2

Reviewer 1 Report

Dear authors,

The submitted manuscript has improved with the latest corrections, although I still see some issues for improvement. I comment on them below:

- What are the future lines of research proposed by the work?
- It is recommended to include tables and / or figures in the presentation of the results to facilitate the reading and interpretation of the data.
- Incorporate and update bibliographic references in the text and at the end of it. Some are recommended:
doi.org/10.3390/su12104176
doi.org/10.3390/su12104091
- Section 3 with its subsections 3.1, 3.2 and 3.3 are too long. You must synthesize them.

Author Response

Dear reviewer!

Thank you for your constructive comments which we found very helpful. We have  incorporated your comments. See attached cover letter.

Best regards

Authors

Reviewer 2 Report

The paper is well written and is consistent with the current state of knowledge related to this topic and has been better substantiated, with a greater bibliographic review, which has been improved, giving it a more international approach, not only in the introduction but also throughout the entire paper, especially in the discussion and conclusions.

As discussed in the limitations of the study, it is a single case study, hence the low number of the sample, which also does not allow us to generalize, but it is valid to know, contextualize and be able to implement strategies and tools in their refugee camp for and for their educational task. You can specify what type of case study it is.

Likewise, it would be appropriate for future research to continue this work to increase the sample considerably.

Regarding the methodology, for future investigations, it is recommended that a protocol be standardized and validated both for field observations and for informal interviews, through some type of rubric or table that allows summarizing the questions and answers, which would allow the analysis to be specified of the data.

The data obtained through the interviews and focus group interviews can be specified in graphs or tables that can visualize the why of the categories obtained and why they have been established.

The discussion is coherent, but in the absence of quantitative data it cannot be properly objectified, as is typical of a case study.

The biggest aspect to improve in the future is to increase the sample, define it specifically and present the data clearly, not just narratively.

It is necessary that they present quantitative results and make use of tables or graphs that evidence or support the conclusions.

It is suggested that they indicate suggestions for improvement to the participants.

Author Response

(The authors gave the same response as above.)

Reviewer 3 Report

Dear Authors,

Thank you for your modifications, which help to improve your manuscript. 

However, new specifications and clarifications are required:

1. In section 2 (line 225 and following), please detail, in greater depth, the sociodemographic description of the participants, including the type of sampling performed. In this subsection, the data collection techniques applied should not be advanced. This description should be made in a new subsection called "data collection techniques or instruments", whose partial information appears in "empirical material", more appropriate to the subsection (also not included) "procedure".

2. Please clarify the data collection techniques finally applied. There seem to be methodological jumps between observation, interview and focus group. Note that there is no technique called "focus group interview". The wording of the indicated subsection would clarify this issue.

3. Subsection 2.6. Bring the limitations of the study to the end of the manuscript.

Author Response

(The authors gave the same response as above.)

Round 3

Reviewer 3 Report

Dear authors,

Thank you for addressing most of our recommendations and suggestions.

Despite the presence of identifiable methodological weaknesses (greater precision in the description of the data collection techniques applied, absence of empirical data of qualitative validity and persistence in maintaining the name of a technique called "Focus group interview"), the interest of the object of study motivates its consideration for publication.

Author Response

Point 1:

Despite the presence of identifiable methodological weaknesses (greater precision in the description of the data collection techniques applied, absence of empirical data of qualitative validity and persistence in maintaining the name of a technique called "Focus group interview"), the interest of the object of study motivates its consideration for publication.

Response 1:

We thank you for your patience and clarifying feedback. We are sorry for our persistence according to the name of the technique focus group interview. We come from four different disciplines and have had discussions related to, among other things, the methodological perspectives.

We have now changed the technique focus group interview to group interview. After considerations and discussions in the research group, we find that group interview is a more precise term. The feedback from the reviewers has made us aware of the complexity of the research field when it comes to group interviews as a method compared to focus groups.

Changes from focus group interview, to group interview are implemented with following changes:

  • The term “focus group interview” are changed to “group interview” throughout the whole article
  • The term is also changed in the table 1.
  • We have added new text in line 230-237 in order to explain the use of group interview as one of the techniques.
  • We have added new bibliography about group interview: [65] Frey, J.H. & Fontana, A. (1991). The group interview in social research. The Social Science Journal, 28(2), 175–187. 
  • We have removed the following bibliography about focus groups [65] Halkier, B. Fokusgrupper.; Gyldendal Akademisk: Oslo, Norway, 2010

All these changes in order to make the empirical research and methods more clearly presented and stated.